# Novel Practical Life Cycle Prediction Method by Entropy Estimation of Li-Ion Battery



**Tae-Kue Kim** [1,*] and **Sung-Chun Moon** [2]

1   Department of Electrical Engineering, Changwon National University, Changwon 51140, Korea
2   Nasan Electric Industries Co., Ltd., Changwon 51124, Korea; sungchun.moon@nasanec.co.kr
*   Correspondence: teakueda@changwon.ac.kr; Tel.: +82-55-213-3630

**Abstract:** The growth of the lithium-ion battery market is accelerating. Although they are widely used in various fields, ranging from mobile devices to large-capacity energy storage devices, stability has always been a problem, which is a critical disadvantage of lithium-ion batteries. If the battery is unstable, which usually occurs at the end of its life, problems such as overheating and overcurrent during charge-discharge increase. In this paper, we propose a method to accurately predict battery life in order to secure battery stability. Unlike the existing methods, we propose a method of assessing the life of a battery by estimating the irreversible energy from the basic law of entropy using voltage, current, and time in a realistic dimension. The life estimation accuracy using the proposed method was at least 91.6%, and the accuracy was higher than 94% when considering the actual used range. The experimental results proved that the proposed method is a practical and effective method for estimating the life of lithium-ion batteries.

**Keywords:** lithium-ion battery; SOL; entropy; PDM; SSM





## 1. Introduction

Li-ion batteries are excellent energy density batteries among the secondary batteries currently being commercialized. Due to their relatively light weight and high energy density compared to other secondary batteries, they are widely used in various fields of technology, from portable products to large energy storage systems. In particular, with the spread of electric vehicles, next-generation batteries such as lithium-sulfur, lithium-air, sodium-magnesium, and solid-state batteries are being developed in order to solve the battery storage capacity and stability issues to improve vehicle mileage. However, these next-generation batteries possess many problems that need to be solved before being commercialized. Therefore, moving forward, the lithium-ion battery market is expected to continue to expand. In particular, with the growth of the mobile phone and personal mobility (electric kickboard, electric bicycle, etc.) market, the problem of efficient energy use and stability of lithium-ion batteries has become increasingly important.

In general, Li-ion batteries are initially stable. However, as the frequency of use increases, the life cycle decreases, and the electrolytes are decomposed due to an oxidation-reduction reaction which forms a solid electrolyte interface (SEI) layer, resulting in increased internal resistance [1–8], thereby, reducing the usable capacity of the battery. The SEI layer acts as a protective film, preventing the electrolytes from continuing to decompose. In terms of entropy, this is a direct cause of the reduction in the battery's reversible capacity. As entropy can be used as an indicator of reduction in battery capacity, research on this has been actively conducted [2,4,6].

In a reduced lifespan, repeated charging and discharging may cause problems such as over-charge, over-discharge, and over-current. In particular, due to the growth of the personal mobility market, with the charging of electric kickboards and electric bicycles inside the home, owners often charge without consideration of lifespan, leading to fire accidents due to overheating and ignition.

Lithium-ion batteries are considered to have reached their end of life when their nominal capacity drops below 80% of their initial capacity [8,9]. Predicting the life of these batteries is very important for efficient resource utilization, stable management, and stable use of energy storage devices. Although much research is being conducted on life expectancy, the representative ones are well summarized in Sim's paper [10]. NASA proposed a life study by measuring the internal impedance of the battery, but this is not suitable for practical applications as it utilized expensive equipment and sophisticated experiments [11]. Widodo proposed using sample entropy, and Chan used an artificial neural network technique to determine the discharge current and capacitance. In addition, according to depth of discharge (DOD), there are capacity estimation methods such as the extended Kalman filter method, and the particle filter method proposed by Plett, however, most methods have a disadvantage in that they cannot cope with the frequently changing use environment caused by using parameters including time functions. In addition, the model estimation method based on the temperature (T) has elements that cause error of the state analysis of the battery due to the boundary condition, the temperature measurement error and the slow dynamic characteristics of the temperature change [12–21]. Therefore, in this paper, we proposed a method to exclude the uncertain existing elements and to apply them based on relationships derived from observations in practice. First, the voltage and state of charge (SOC) relationship was identified from an entropic point of view to eliminate the time factor by considering the actual battery management system configuration. Next, we presented a method for predicting battery life by using the derived relationships. Finally, we validated the proposed method by minimizing the calculation error and the life expectancy error in the process of calculating irreversible energy.

## 2. Suggestion of a Model for Predicting Battery Lifetime

### 2.1. The Charging-Discharging Characteristics of Li-Ion Batteries

Based on the Shepherd model, the characteristic equation of the voltage for charging and discharging of the battery can be expressed as Equations (1) and (2) with parameters related to current, temperature, and state of health (SOH) [1].

$$E_t = E_0 - R_i \cdot I - K\frac{Q_0}{I \cdot t - 0.1Q_0} \cdot I^* - K\frac{Q_0}{Q_0 - I \cdot t} \cdot I \cdot t + Ae^{-(B \cdot I \cdot t)} \quad (I^* < 0) : Charging \quad (1)$$

$$E_t = E_0 - R_i \cdot I - K\frac{Q_0}{Q_0 - I \cdot t} \cdot I^* - K\frac{Q_0}{Q_0 - I \cdot t} \cdot I \cdot t + Ae^{-(B \cdot I \cdot t)} \quad (I^* > 0 : Discharging) \quad (2)$$

Equations (1) and (2) are voltage characteristic equations for battery analysis, which many researchers present as a basic model. However, in this case, even if the current is interpreted as a constant current (CC) operated by the CC mode, $E_0$, $R_i$, $K$, $Q_0$ are part of a function that can be expressed as Equations (3)–(6) according to temperature and SOH.

$$E_0 = E_{0|T_n} + \frac{\partial E}{\partial T}(T_i - T_n) \tag{3}$$

$$R_i = R_{i|T_n} \cdot \exp\left(\beta\left(\frac{1}{T_i} - \frac{1}{T_n}\right)\right) \tag{4}$$

$$K = K_{|T_n} \cdot \exp\left(\alpha\left(\frac{1}{T_i} - \frac{1}{T_n}\right)\right) \tag{5}$$

$$Q_0 = Q_{0|T_n} + \frac{Q_0}{T}(T_a - T_n) \tag{6}$$

Therefore, it is difficult to analyze the characteristics of the battery by only using Equations (1) and (2). In addition, since the above equations are calculated as a function of time, it may be an unrealistic analysis method because there are many errors due to changes in the usage pattern of the battery or the load environment. In addition, estimating the internal resistance, polarization resistance, and capacity reflecting thermal factors, which

are analyzed in many studies, actually include many error factors that cannot be used in the analysis. In order to prevent this problem, a method of analyzing the state by measuring the state of the current "0" has been proposed, however, it takes some time to stabilize before measuring the open circuit voltage (OCV). Therefore, it is difficult to apply to the real-time prediction method. Considering the realistic conditions of use, the actual measurable parameters are voltage, current, and temperature. Temperature is an important parameter here, however, in high-capacity applications such as for an energy storage system (ESS), a cooling system is provided for stable battery operation, and a system is operated to control the battery surface temperatures. However, most small-capacity systems do not have cooling facilities, do not maintain an adequate I-rate, and the temperature is only used as a protective element by limiting the maximum temperature. When considering the heat transfer characteristics inside and outside the battery, and the slow dynamics of the temperature measurement, it can be easy to misinterpret the conditions of the battery when reflecting them in real-time calculations.

### 2.2. The Mathematical Model for Suggested Life Cycle Prediction

To establish optimal operating conditions for battery performance, such as battery life and safety, it is important to quantify heat generation and temperature changes based on the C-rate of charge/discharge.

$$G(x) = -nFE_{oc}(x) = H(x) - TS(x) \tag{7}$$

The equation for enthalpy ($\Delta H$) and entropy ($\Delta S$) is explained in terms of Gibb's free energy ($\Delta G$), and is shown in Equation (7), where $n$ is the number of electrons included in the reaction ($n = 1$ for lithium ions), and $F$ is a Faraday constant. Here, $x$ represents the concentration of lithium ions, and since this value is proportional to SOC, it can be expressed as Equation (8).

$$S(x) = F\left(\frac{\partial E_{oc}(x)}{\partial T}\right) = F\left(\frac{\partial E_{oc}}{\partial T}\right)_{SOC} \tag{8}$$

Based on the above equation, we can define the irreversible joule heat, the reversible joule heat, and the heat generated by the terminal resistance, as shown in Equation (9) [2].

$$\dot{Q}_{total} = \dot{Q}_{irreversible} + \dot{Q}_{reversible} + \dot{Q}_{tab} = I(E_{OC} - E_t) - IT\left(\frac{\partial E_{OC}}{\partial T}\right)_{SOC} + I^2(R_A(T) + R_C(T)) \tag{9}$$

Here, we focused on irreversible joules because in the case of functions related to temperature or internal resistance, as mentioned above, errors are more likely to occur due to measurement or estimation errors and slow dynamics. Therefore, we proposed a method of applying irreversible energy in terms of voltage and current which can be estimated in real-time.

$$\dot{Q}_{irreversible} = \dot{Q}_{ir} = I(E_{OC} - E_t) \tag{10}$$

Here, $E_{OC} = \alpha E_C$ and $E_t = E_D$ can be substituted, where $E_C$ represents the cell voltage during charging and $E_D$ represents the cell voltage during discharge. The total amount of irreversible energy can be expressed as Equation (11) by integrating Equation (10).

$$Q_{ir} = I\int(\alpha E_C - E_D)dt \tag{11}$$

The SOC may be expressed with respect to the current battery state in relation to the total battery capacity, as shown in Equation (12).

$$SOC = \frac{Q_b \pm Q_t}{Q_0} = (Q_b \pm I{\cdot}t)Q_0^{-1} \tag{12}$$

Taking the derivative in Equation (12), it can be rearranged as in Equation (13).

$$dt = I^{-1} \cdot Q_0 dSOC \tag{13}$$

Taking Equation (13) forward, both sides are multiplied by $V_b$, before converting it into an integral, as expressed in Equation (14).

$$\int V_b dt = I^{-1} \cdot Q_0 \int V_b dSOC \tag{14}$$

We defined $V_b$ as being $\alpha E_C - E_D$, therefore, it can be expressed as Equation (15).

$$\frac{1}{Q_0} I \int (\alpha E_C - E_D) dt = \int (\alpha E_C - E_D) dSOC \tag{15}$$

If Equation (15) is re-arranged from Equation (11), it can be seen that the total amount of voltage change for SOC can be derived in such a way that irreversible energy can be obtained, as seen in Equation (16).

$$\frac{Q_{ir}}{Q_0} = \int (\alpha E_C - E_D) dSOC \tag{16}$$

For each charge and discharge cycle, the life of the battery can be calculated by adding the irreversible energy ($Q_{ir\_k}$) and comparing it with the maximum cycle product of the maximum amount of irreversible energy ($Q_{ir\_m}$) that can be generated in one cycle. Using Equation (17), the state of life (SOL) can be estimated. Therefore, the life of the battery can be determined by calculating the irreversible energy according to the charging and discharging. In general, it is considered to have reached the end of its life when it reaches 80% of its initial battery capacity or less [12].

$$SOL = 1 - \frac{1}{mQ_{ir\_m}} \sum_{k=1}^{n} Q_{ir\_k} \tag{17}$$

Since the charge and discharge cycle provided by the battery manufacturer requires a reference number at 100% of the DOD, the actual usable charge and discharge cycle may be calculated through Equation (18).

$$\frac{\sum_{k=1}^{n} Q_{ir\_k}}{\min(mQ_{ir\_m}, \sum_{k=1}^{n} Q_{ir\_m})} = \frac{N_a}{N_p} \tag{18}$$

## 3. Experiments and Discussion

### 3.1. Configuration of Test System

To verify the proposed method, the battery charge/discharge test system was constructed as shown in Figure 1. At first, we proceeded to turn the system on/off according to the charging/discharging sequence in the main controller by using a battery charger with CC and CV functions. The operation of CV with CC could be implemented by using a power supply with a current limit function. The switch and the charging/discharging sequence were controlled by communication. The temperature T is configured to operate only to protect against abnormal conditions. We constructed a system that receives voltage, current, time, and SOC information indicating the state of the battery. Also, we can configure it to control the charge/discharge mode and settings using a human machine interface (HMI), and the system can be analyzed for battery state, SOL, and SOH. The test system had the function to compare and analyze the data received from the BMS and the measured data to identify its reliability.

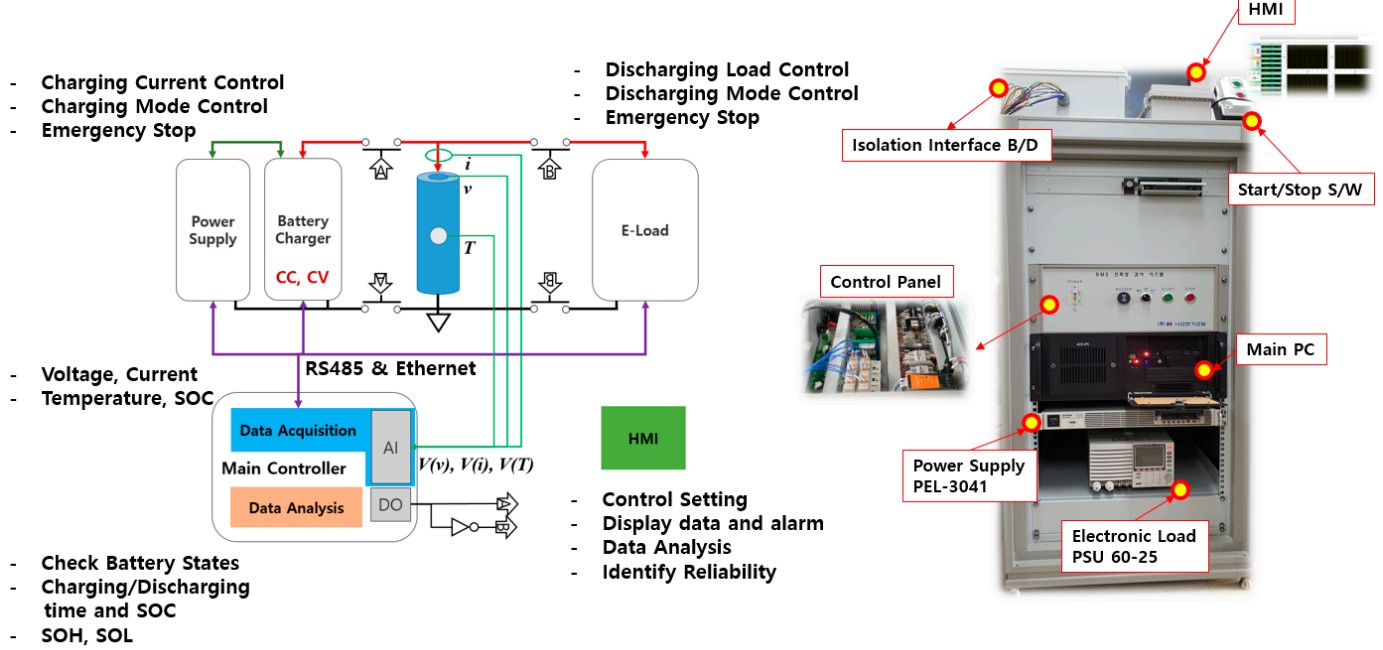

**Figure 1.** Configuration of the battery cycle test system.

### 3.2. Characteristic Test of Li-Ion Batteries

The Li-ion battery used in this experiment, which was newly installed into the company A battery bank of company A, is outlined in Table 1 (note that it does not refer to a specific company's battery).

**Table 1.** Specification of the Li-ion battery used as a test model.

| Parameter | Value | Parameter | Value |
|---|---|---|---|
| Standard discharge capacity, mAh | min 4900 | Cycle life capacity at 500 cycles, mAh | ≥3802 |
| Rated discharge capacity, mAh | min 4753 | Initial internal impedance, mΩ | 28.0 |
| Charge voltage, V | 4.5 | Calculated internal impedance, mΩ | 40.0 |
| Nominal voltage, V | 3.63 | Cell weight, g | 69.0 |
| Max. charge current, mA | 4900 | Cell length, mm | 70.6 |
| Max. cont. discharge current, mA | 9800 | Cell diameter, mm | 21.1 |
| Discharge cut-off, V | 2.5 | Charge method | CC-CV [1] |

[1] Constant voltage with limited current.

To investigate the characteristics of the aging state of the battery, respectively characteristic experiments were conducted on six samples for which the batteries have run for 0, 100, 200, 300, 400 and 500 cycles with the rated charging current of 0.5 C and the discharge current of 1 C. The DOD was 100 percent. Figure 2a shows a graph of discharge characteristics when discharged at 1 C for the battery at different states. Figure 2b shows charging characteristics when charged at 0.5 C. As we would expect, the more the charge and discharge cycle was, the faster the charge and discharge rate was. This is because the charging and discharging was performed according to a precisely defined rule for battery characteristics, and thus, the characteristics of each aging state showed a very linear change.

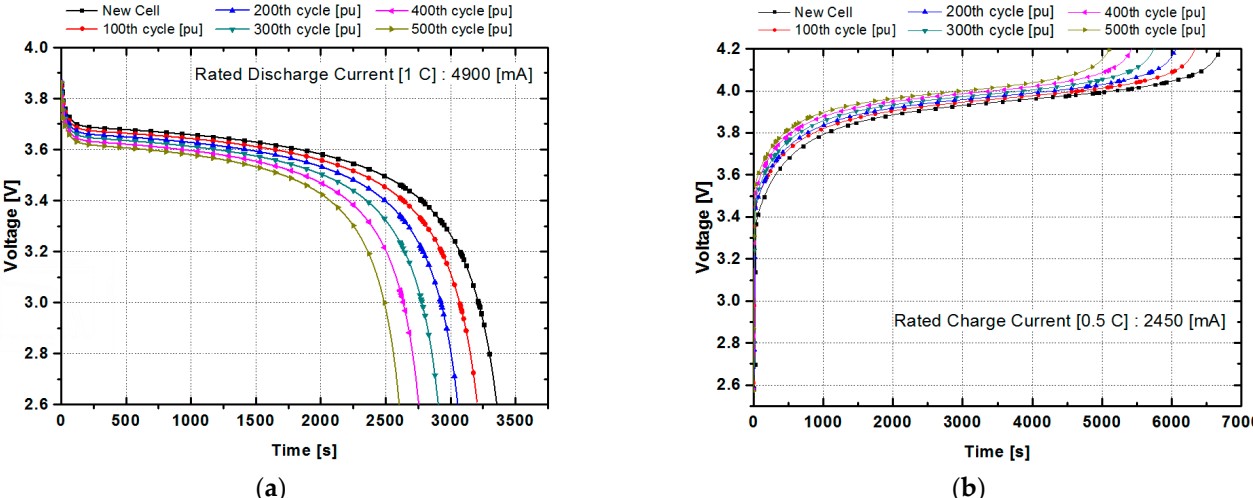

**Figure 2.** The charging and discharging characteristic curves of the lithium-ion battery: (**a**) battery characteristics using a 1 C discharging current according to the battery aging condition (cycle number) and (**b**) battery characteristics using a 0.5 C charging current according to the battery aging condition (cycle number).

Figure 3a is a graph of voltage against SOC when the discharge current is fixed at 1 C, the rated current, and the charge current is discharged at 100% for different charge currents ranging between 0.1 C and 1 C. It can be seen that the area of the curve that appears during charging and discharging changes with the magnitude of the current. In the case of 1 C, which has the largest current, the area is large, and it decreases as the charging current decreases. Figure 3b shows the graph of voltage against SOC while changing the discharge current to 100% DOD with the charging current fixed at 0.5 C. In the case of 1 C, which was the largest discharge current, it can be seen that a line is drawn at the bottom of the graph. As the current decreases, the area covered by the graph decreases. Figure 3a,b confirms the rationale for the method presented above. Furthermore, it can be seen that the area surrounding the curve of voltage against SOC changes according to the magnitude of the current, and the area increases as the current increases. This amount can be converted into irreversible energy, and the life cycle can be estimated using the method of calculating the irreversible energy presented in this paper.

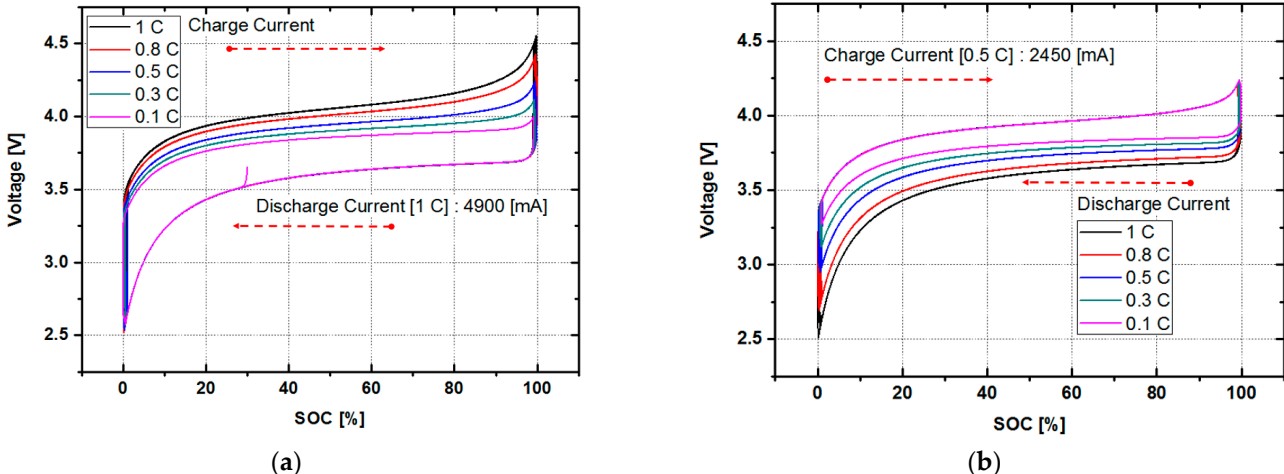

**Figure 3.** The voltage curves for SOC using different currents: (**a**) voltage against SOC when charging with different currents. The result using a fixed discharge current of 1 C is also shown and (**b**) voltage against SOC when discharging at different currents. The result using a fixed charge current of 0.5 C is also shown.

Figure 4a shows the charge/discharge characteristic curves at 100% DOD for 0.5 C charging (0.5 CC) & 1 C discharging (1 CD) and 0.1 C charging (0.1 CC) & 0.1 C discharging (0.1 CD). The area of the curved surface surrounded by the solid black line in Figure 4a is defined as $Q_{ir\_m}$ and the rated charge current and rated discharge current for one cycle is the maximum size of irreversible heat capacity when operated at 100% DOD. The area represented by the solid red line is the magnitude of the irreversible heat capacity at 100% DOD for 0.1 C charging (0.1 CC) & 0.1 C discharging (0.1 CD), defined as $Q_{ir\_k}$, meaning the attenuated irreversible heat capacity of the battery. Figure 4b shows the capacity consumed when the DOD is operated at 70%. In Figure 4a,b, $Q_{ir\_m} - Q_{ir\_k}$ defines the remaining available capacity. Through this, by designing a proportional relationship, we can calculate the remaining capacity and the remaining cycles.

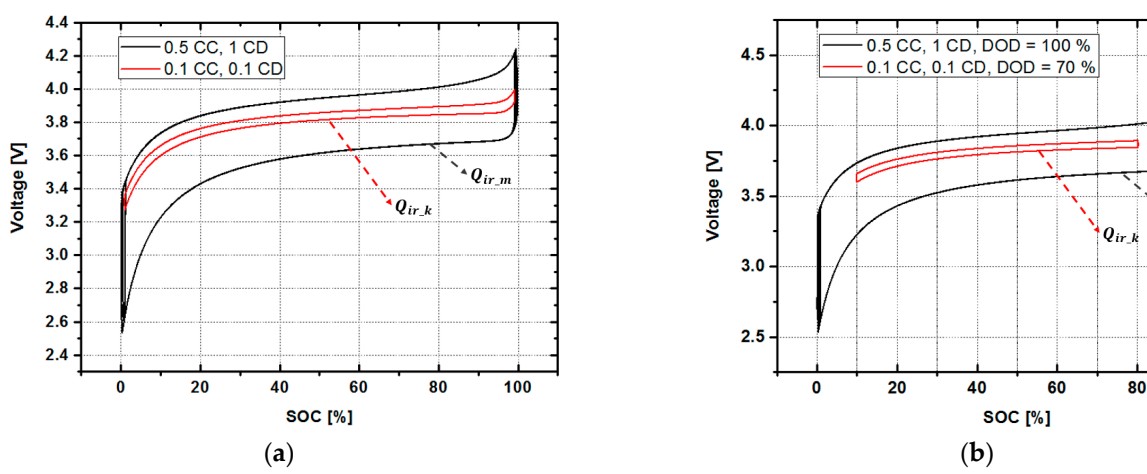

**Figure 4.** Change in the heat energy (Q) value according to the situation: (**a**) when only the charge/discharge currents are different and (**b**) when the charge/discharge currents are different and discharge of depth (DOD) is different.

Figure 5 shows how varying the magnitude of the charge-discharge current and the DOD can alter results. The black solid line region is the reference $Q_{ir\_m}$, which is the maximum irreversible energy in one cycle. This graph shows the shape of the irreversible heat capacity when changing the charging/discharging characteristics. If the C-rate of the current exceeds the rated range, the irreversible heat capacity exceeds the one-cycle reference value, which shortens battery life. Charge-discharge behavior below the rated current means less irreversible energy, which in turn means increased residual life.

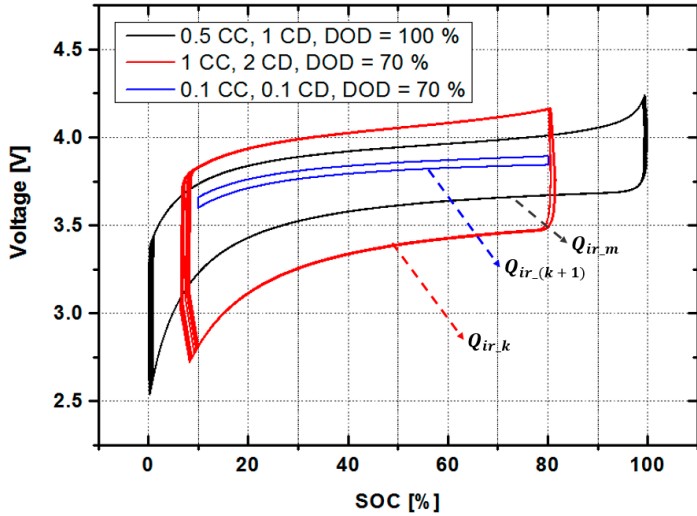

**Figure 5.** The irreversible energy in various situations.

Figure 6a shows a comparison of the characteristic curves when batteries with six different lifetimes were charged and discharged at 0.5 CC, 1 CD, and DOD = 100%. The area within the curve of the new battery is the smallest, and as life is shortened, the area of the curve increases. Figure 6b shows the characteristic curves of a new battery and one that is on its 500th cycle. When a battery has reached the end of its life, it is discharged until the SOC reaches 0%, resulting in a voltage drop phenomenon occurring. At this point, it is expected that battery problems will occur due to instantaneous transients.

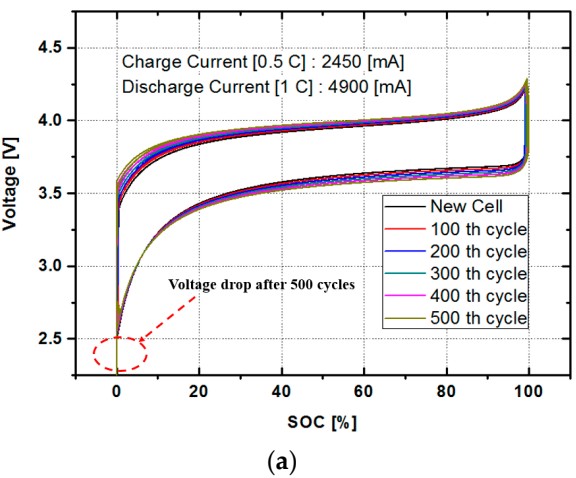 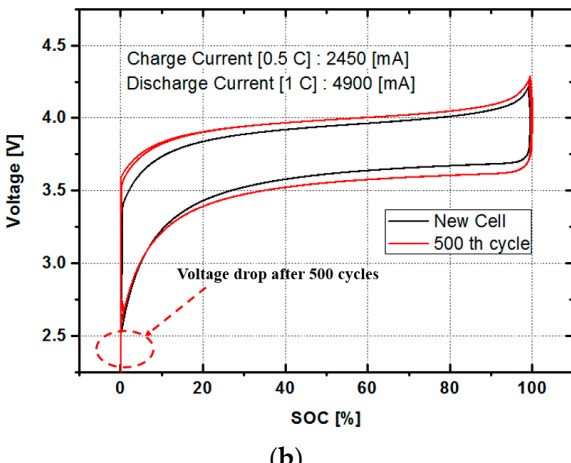

(**a**)          (**b**)

**Figure 6.** The voltage curve for SOC with different aging conditions (cycle number) using the same charge/discharge current: (**a**) voltage against SOC when using 0.5 constant current (CC), 1 discharge current (CD) for different aging conditions (0–500th cycle) and (**b**) voltage against SOC when using 0.5 CC, 1 CD for a new battery and a battery on its 500th cycle.

### 3.3. Signal Processing Strategy

### 3.3.1. Point Detection Method (PDM)

In general, you use the integrator to find the area of the curve. However, the integrator has a drawback in that it must be processed by applying the method for removing accumulated error. In this paper, we applied the method of finding the area of a rectangle by detecting four inflection points without using an integrator. Using this method, we can eliminate the error of using an integrator, and in a simpler and more efficient way, we can obtain the irreversible heat capacity ($Q_{ir\_k}$) that we are looking for.

Figure 7 shows the four-point changes that need to be acquired. The point change is divided in Figure 7a when the discharged energy is smaller than the charged energy, and in Figure 7b when the discharged energy is larger than the charged energy. $P_{CS}$ is defined as the charging start point, $P_{CK}$ as the charging knee point, $P_{DS}$ as the discharging start point, and $P_{DK}$ as the discharging knee point. This change in point can follow a total of four different modes, which are shown in Figure 8.

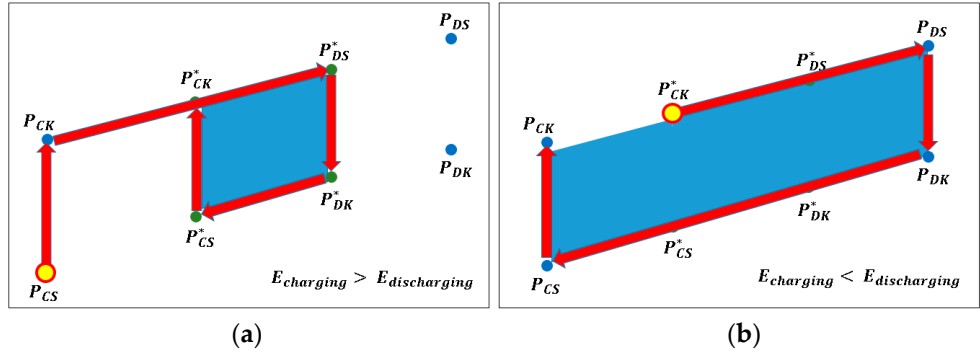

(**a**)          (**b**)

**Figure 7.** The point detection method: (**a**) when charging energy is more than discharging energy and (**b**) when the charging energy is less than the discharging energy.

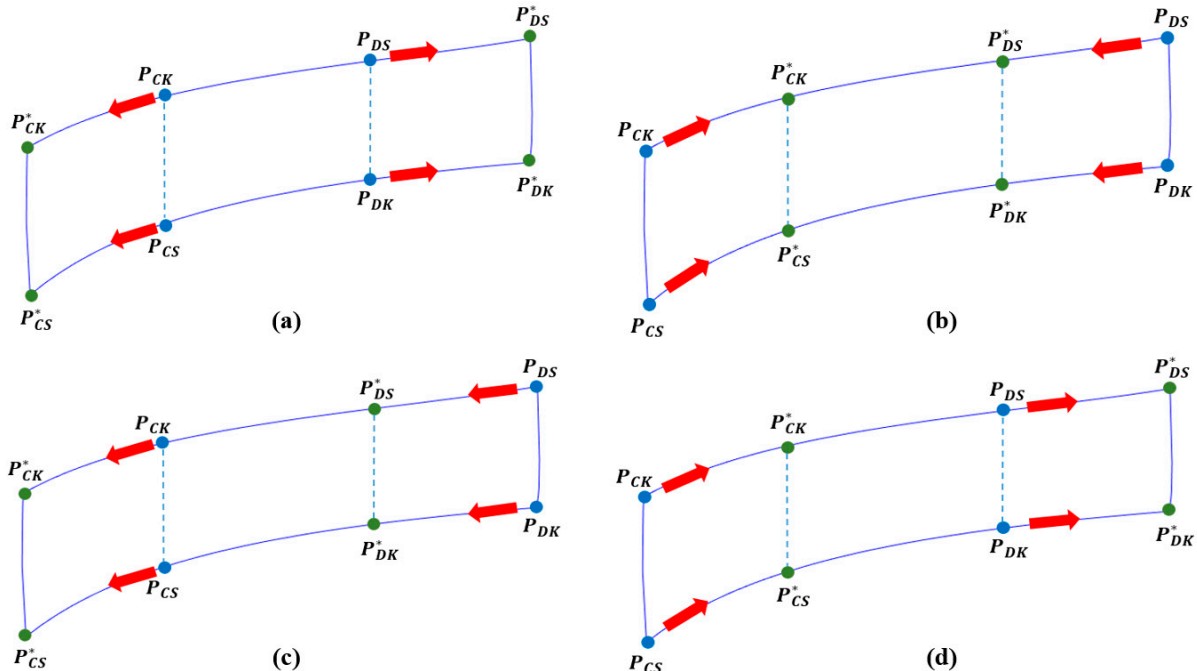

**Figure 8.** Four modes of the point detection method (PDM) according to charge and discharge status: (**a**) if the DOD is widened; (**b**) if the DOD is narrowed; (**c**) if the DOD range of movement is low; and (**d**) if the DOD range of movement is high.

Figure 8 can be expressed as a DOD. In the case of Figure 8a, it means that the range of DOD is wider than before. In the case of Figure 8b, the range of DOD is narrowed. In the case of Figure 8c, the DOD area is moved, that is, the depth of discharge increases, and the depth of charge decreased. In this case, if the depth of charge is also high, then it becomes the same as in Figure 8a. On the contrary, in Figure 8d, the depth of discharge becomes shallower and the depth of charge increases. Similarly, if the charging depth is also decreased, then it becomes the same as in Figure 8b. In general, most of the charge/discharge states of a battery are included in the four cases above. In Figure 7, the blue marked $P$ ($P_{CS}$, $P_{CK}$, $P_{DS}$, $P_{DK}$) represents the current point and is equal to $P(n-1)$ of the discrete signal processing component. The green point, represented by the superscript, $P^*$ ($P^*_{CS}$, $P^*_{CK}$, $P^*_{DS}$, $P^*_{DK}$) is the newly updated point, which is $P(n)$ of the discrete signal processing component.

3.3.2. Section Separation Method (SSM)

The area can be obtained by using the above PDM, however, when looking at the shape of the curve, a wider DOD increases the error in area calculation as shown in Figure 8a. Of course, the actual use range for most batteries is 20–80%, meaning there is no big error because the curve is an almost constant rectangular area. However, when considering the whole area of SOC (0–100%), this method may be unsuitable. Therefore, as shown in Figure 8b, we propose the SSM method by separating the 3 sections. Table 2 indicates the accuracy rate of the data calculated by PDM for the actual curve area. If SSM is not applied, it can be seen that the accuracy rate significantly decreases in the large DOD period.

**Table 2.** The accuracy rate with calculated method.

| DOD [1] [%] | No Section [%] | 3 Section Separation [%] |
|---|---|---|
| 0–100 | <70 | >89 |
| 10–90 | >88 | >92 |
| 20–80 | >92 | >94 |

[1] DOD: depth of discharge.

### 3.3.3. Algorithm

The practical algorithm to implement the method presented in this paper is shown in Figure 9. First, to determine $Q_{ir\_m}$, either the manufacturer's provided information was used or a sample charge/discharge cycle was executed before storing the $Q_{ir\_m}$ information. After that, the voltage, current, and temperature information were acquired in real-time, and the information was used to predict the lifetime. The temperature information was used as an emergency stop trigger for abnormal situations. Using the above information, first, check whether it is in a state of charge or discharge. When the direction of the current is changed by using the current information, it is the information of the point of starting charging or discharging, and thus storing it in $P_{CS}$ and $P_{DS}$. The SOC information is then checked during each stage of charging and discharging. When the SOC at the start of charging or discharging and the current SOC differ by more than 1%, the point is determined as a knee point and the voltage information at the knee point is stored in $P_{CK}$ or $P_{DK}$. As shown in Figure 10, the area for the four points becomes a value that be used to calculate the irreversible energy, so the areas can be calculated by Bretschneider's or Brahmagupta's formula. This cycle is divided into sections based on SOC information, and the same operation is performed in three sections, and this information is used to calculate a value corresponding to the irreversible energy of one cycle and is estimated by applying Equations (17) and (18) to predict the lifetime.

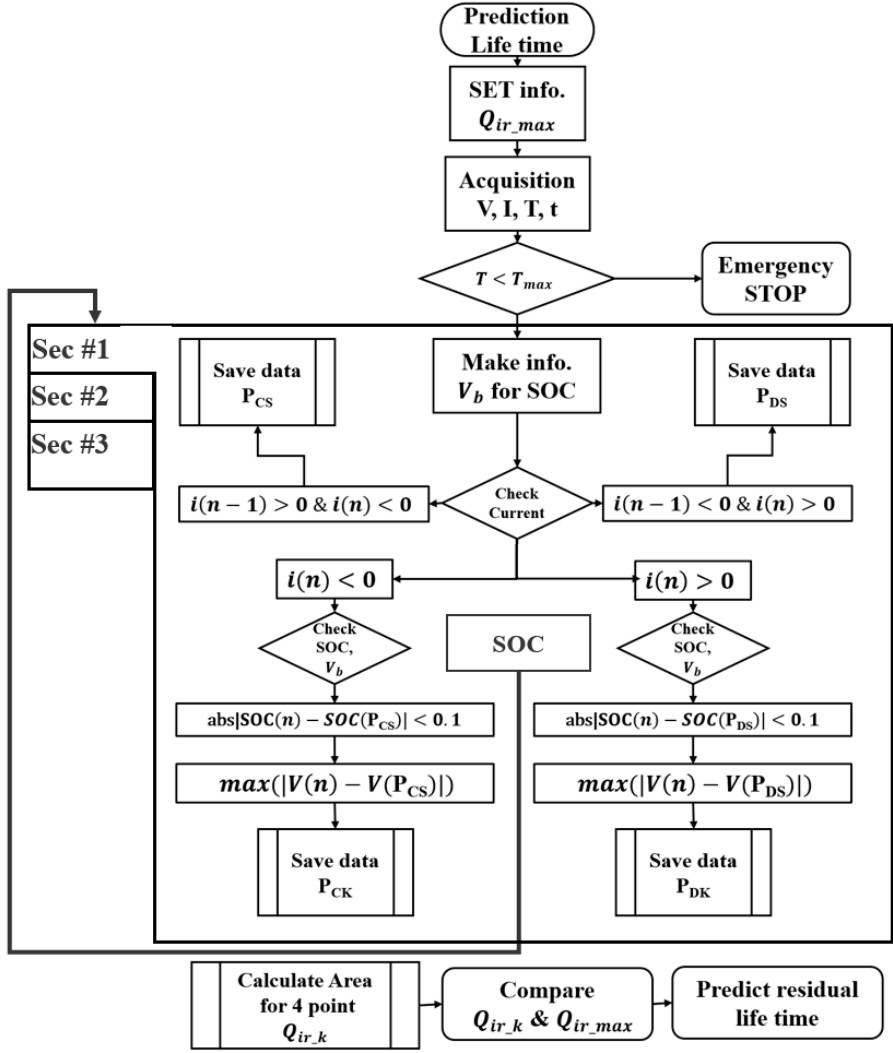

**Figure 9.** The algorithm for predicting residual lifetime.

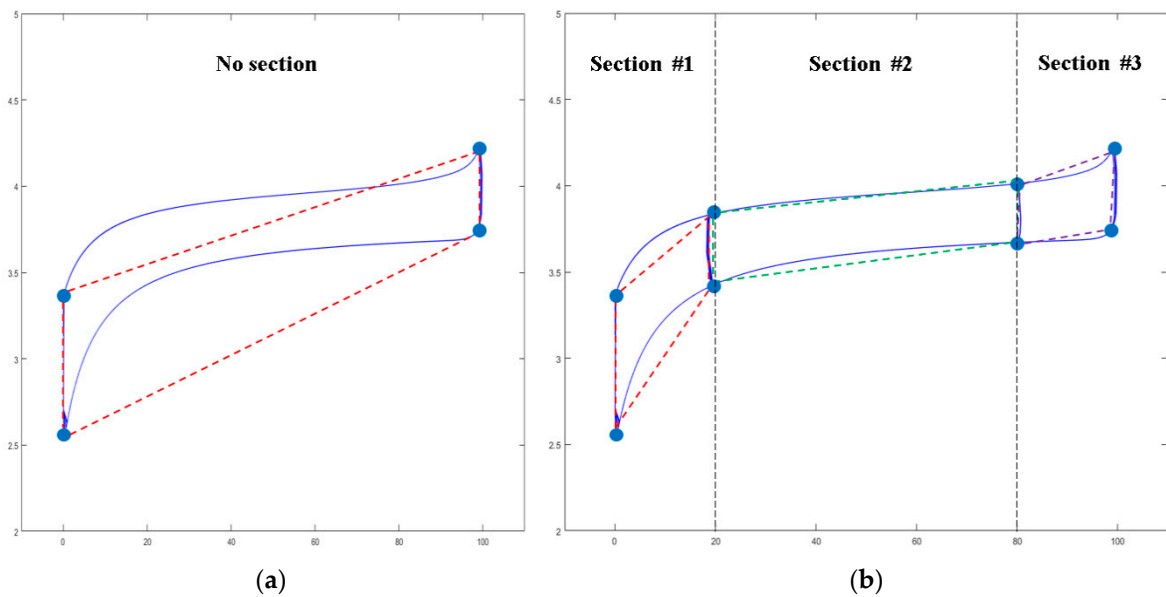

(**a**)　　　　　　　　　　　　　　　　　　　　　　　　　　　　(**b**)

**Figure 10.** The concept of applying the section separation method: (**a**) the estimated Q value by PDM if the section separation method (SSM) is not applied and (**b**) the estimated Q value by PDM if SSM is applied.

## 4. Results and Conclusions

Figure 11 shows the comparison of the voltage change for time and the voltage change for SOC when the discharge current is different according to real-time change. Figure 11a shows that the state of voltage changes as the magnitude of discharge current changes to 0.5 C, 0.25 C, and 1 C with a time interval. However, in Figure 11b, although the discharge current changes with a time interval, the voltage change for SOC shows a constant cycle curve regardless of the time change. From this, it can be seen that it is a very effective method of predicting lifetime by using the voltage/SOC curve regardless of the change in the user's charge/discharge pattern. Here, considering a more realistic battery usage method, we verified the performance by applying the proposed method to a 12s-1p (12s: 12 series, 1p: 1 parallel) battery module with BMS. Figure 11c shows that the state of voltage changes of the 12s-1p battery module and Figure 11d shows channel 1's voltage and current variations.

Table 3 shows the results of comparing and analyzing the predicted battery life and the actual remaining battery capacity by applying the method proposed in this paper, while changing the magnitude of the battery charge-discharge current and DOD. As a result, it was confirmed that there is some difference in estimation accuracy for the different DODs. The accuracy was slightly lower in the 0–100% range, and the smaller the DOD was, the lower the estimation error was, leading to higher accuracy. The estimated accuracy deviation in the 100% DOD section and the 50% DOD section differed by a maximum of 2.7%, and it can be seen that a deviation of about 1% occurred depending on the magnitude of the charge/discharge current.

Figure 12 is a graphical representation of the data shown in Table 3. It can be seen that the remaining life greatly depends on the charge-discharge C-rate and the DOD. The results of the predicted lifespan were compared based on the actual remaining capacity, and the predicted lifespan was calculated to have an average accuracy of 93% or more. In the high DOD section, the accuracy was somewhat reduced, and the 20–80% section and 30–80% section, which are the expected use sections, had an accuracy of over 94%. Considering the analytical and experimental results above, it can be seen that first of all, DOD and C-rate are responsible for the lifetime. As can be seen from the analysis results on the curve for calculating the irreversible heat capacity presented in this paper, the area of the Q value is clearly different according to the DOD and the C-rate. As a result, the larger the DOD was, the larger the C-rate was, and the higher the Q value was, which means that

the lifetime was shortened. Rather, under the same conditions (same DOD and C-rate), irreversible heat capacity increased according to the aging state of the battery, but the rate of aging was relatively lower than the effect of the DOD or the C-rate.

As a result of the experiment, it can be seen that the temperature (T) slightly influenced the results. It can be seen that when the temperature was out of the proper operating range, the calculated irreversible energy increased, meaning a shortening of the lifetime. From Equation (18), it can be found that since the numerator term was increased, the predicted lifetime yields a relatively small value. Similarly, as the battery ages, the calculated irreversible energy increases. This part is also expected to produce the same effect. The results presented by this method in this paper show an average life expectancy accuracy of more than 90%, but in order to increase the accuracy, the $Q_{ir\_m}$ value should be reflected according to temperature and aging conditions. Considering the safety factor of the battery system, the minimum $Q_{ir\_m}$ was chosen as shown in Equation (18) to reduce the error for relationships that have not yet been identified. We know that irreversible heat energy increases with lifetime, as shown in Figure 6. However, research is still insufficient to generalize.

**Table 3.** The comparison of the predicted cycle and the actual cycle.

| Charging Current [C] | Discharging Current [C] | DOD [%] | Actual Cycle | Predicted Cycle | Accuracy Rate [%] |
|---|---|---|---|---|---|
| 0.5 | 1 | 100 [0–100] | 554.5 | 510 | 92.0 |
| 0.5 | 1 | 80 [10–90] | 704.0 | 650 | 92.4 |
| 0.5 | 1 | 70 [20–90] | 841.3 | 778 | 92.5 |
| 0.5 | 1 | 60 [20–80] | 983.1 | 918 | 93.4 |
| 0.5 | 1 | 50 [30–80] | 1238.8 | 1172 | 94.6 |
| 0.25 | 1 | 100 [0–100] | 583.1 | 538 | 92.2 |
| 0.25 | 1 | 80 [10–90] | 736.5 | 685 | 93.0 |
| 0.25 | 1 | 70 [20–90] | 879.3 | 815 | 92.7 |
| 0.25 | 1 | 60 [20–80] | 1024.4 | 954 | 93.1 |
| 0.25 | 1 | 50 [30–80] | 1290.8 | 1218 | 94.4 |
| 0.1 | 1 | 100 [0–100] | 633.3 | 585 | 92.3 |
| 0.1 | 1 | 80 [10–90] | 795.3 | 740 | 93.1 |
| 0.1 | 1 | 70 [20–90] | 950.7 | 890 | 93.6 |
| 0.1 | 1 | 60 [20–80] | 1099.3 | 1031 | 93.8 |
| 0.1 | 1 | 50 [30–80] | 1374.4 | 1299 | 94.5 |
| 0.5 | 2 | 100 [0–100] | 381.0 | 350 | 91.8 |
| 0.5 | 2 | 80 [10–90] | 476.4 | 443 | 92.9 |
| 0.5 | 2 | 70 [20–90] | 568.6 | 532 | 93.5 |
| 0.5 | 2 | 60 [20–80] | 663.6 | 622 | 93.8 |
| 0.5 | 2 | 50 [30–80] | 837.4 | 787 | 94.0 |
| 0.5 | 0.5 | 100 [0–100] | 822.7 | 765 | 93.0 |
| 0.5 | 0.5 | 80 [10–90] | 1045.2 | 979 | 93.7 |
| 0.5 | 0.5 | 70 [20–90] | 1251.6 | 1175 | 93.9 |
| 0.5 | 0.5 | 60 [20–80] | 1459.6 | 1376 | 94.3 |
| 0.5 | 0.5 | 50 [30–80] | 1843.9 | 1744 | 94.6 |
| 0.5 | 0.1 | 100 [0–100] | 2078.5 | 1933 | 93.0 |
| 0.5 | 0.1 | 80 [10–90] | 2612.8 | 2453 | 93.9 |
| 0.5 | 0.1 | 70 [20–90] | 3120.3 | 2936 | 94.1 |
| 0.5 | 0.1 | 60 [20–80] | 3607.9 | 3406 | 94.4 |
| 0.5 | 0.1 | 50 [30–80] | 4553.7 | 4308 | 94.6 |
| 1 | 2 | 100 [0–100] | 366.8 | 336 | 91.6 |
| 1 | 2 | 80 [10–90] | 452.3 | 418 | 92.4 |
| 1 | 2 | 70 [20–90] | 548.7 | 510 | 92.9 |
| 1 | 2 | 60 [20–80] | 640.9 | 597 | 93.1 |
| 1 | 2 | 50 [30–80] | 810.4 | 756 | 93.3 |

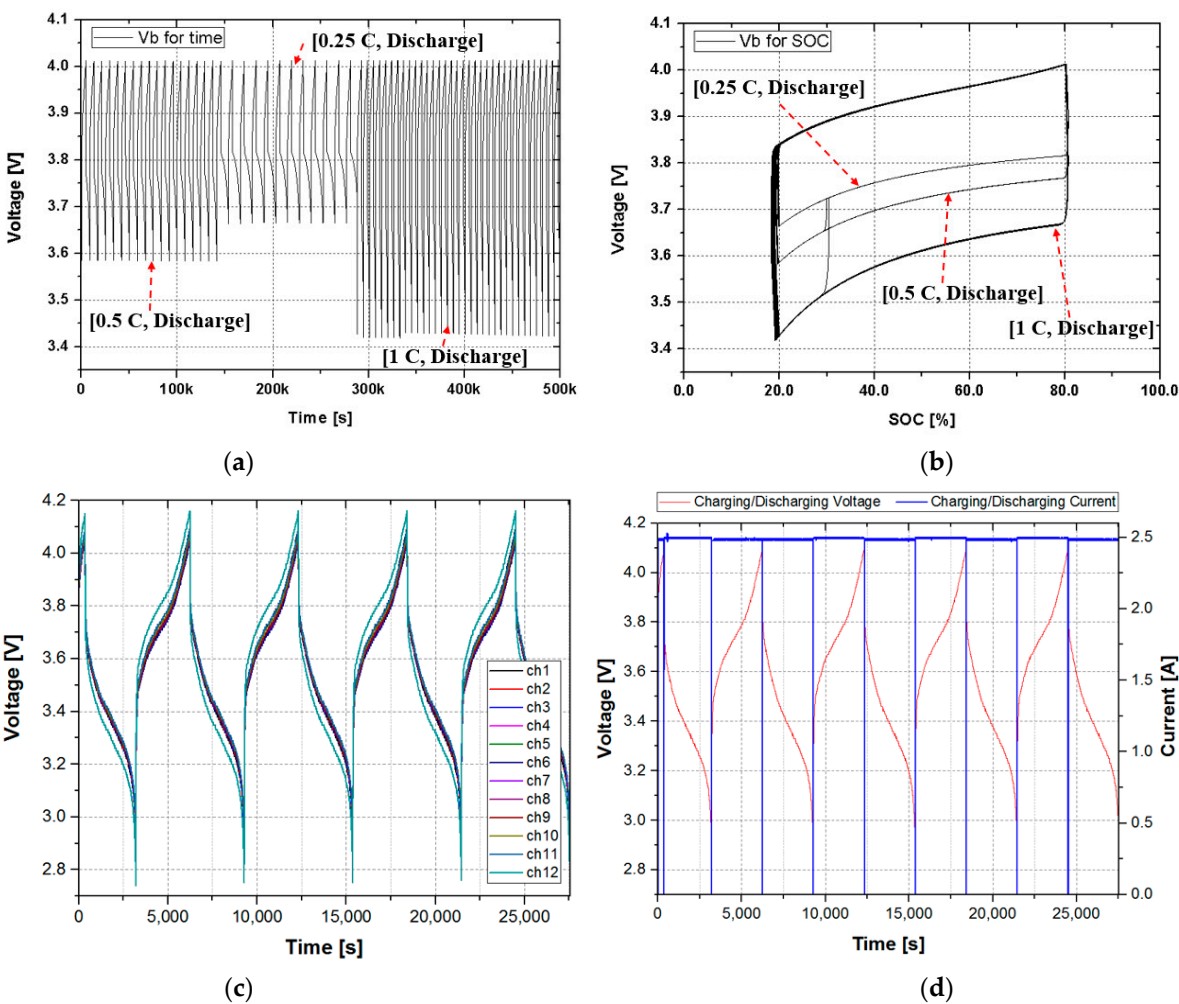

**Figure 11.** The comparison of the voltage against time and the voltage against SOC according to the discharge current change: (**a**) graph of the voltage change against time at 0.25 C, 0.5 C, and 1 C; (**b**) graph of voltage change against SOC at 0.25 C, 0.5 C, and 1 C; (**c**) graph of voltage change for 12 cells; (**d**) and graph of voltage change and current for channel 1.

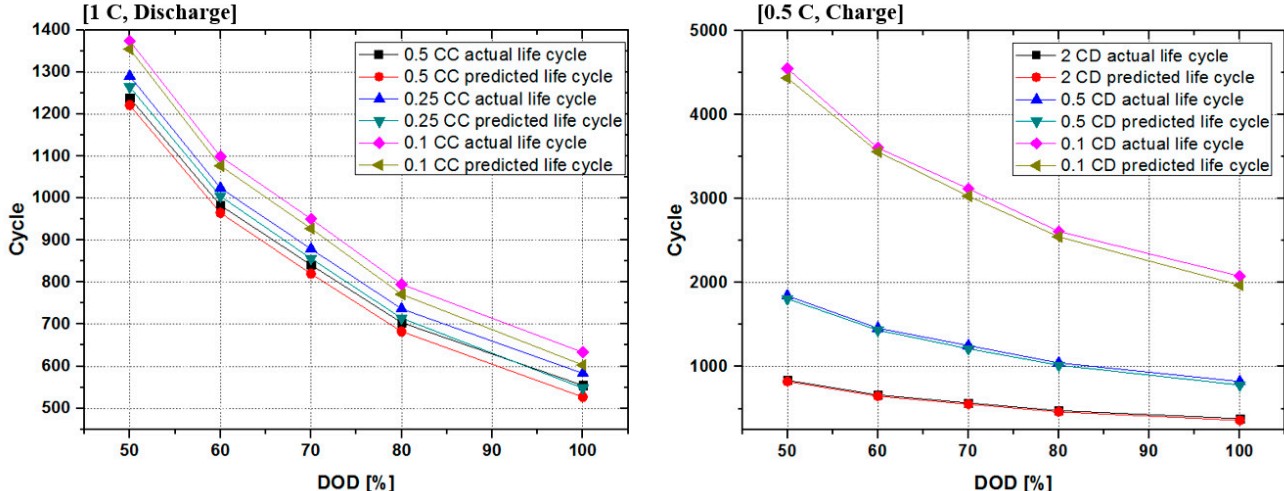

**Figure 12.** The comparison curve of the predicted life cycle and the actual life cycle.

In the present paper, we proposed a practical and effective method of predicting the life of lithium-ion batteries based on the entropy law and we verified the results. The proposed method can be implemented in an intuitive and relatively easy way, while still being a physical system. As mentioned in the introduction above, the estimation and interpretation methods involving the functions of temperature and time are likely to contain errors in their results. In fact, it was noted that it is more effective to judge using the information of voltage and current which are more responsive than the temperature in the process of acquiring and processing the state of the battery in real-time. The voltage reflects the internal state, temperature, and environmental factors of the battery, making analysis easier to access. In addition, by using the voltage for SOC information without using a component that changes with time, the influence on the time component that occurs according to the usage pattern is removed. Furthermore, the experimental results through the proposed life estimation method proved that the proposed method was valid and correct by confirming that the estimated life and the actual life were very similar, with an accuracy of over 92%. In the future, we will study ways to compensate for irreversible capacity increases as the lifespan decreases, and methods on ways to improve life prediction accuracy.

**Author Contributions:** Conceptualization, methodology, software, formal analysis, original draft preparation, T.-K.K.; validation, resource, S.-C.M. All authors have read and agreed to the published version of the manuscript.

**Funding:** This research received no external funding.

**Conflicts of Interest:** The authors declare no conflict of interest.

## Nomenclature

| | | | |
|---|---|---|---|
| $E_t$ | nonlinear battery voltage, V | $\alpha$ | Arrhenius rate constant for $K$ |
| $E_0$ | constant voltage, V | $E_{oc}$ | open circuit voltage, V |
| $R_i$ | internal resistance, $\Omega$ | $Q_{ir}$ | irreversible heat energy, $Q_{irrevesibe}$, W·s |
| $K$ | polarization constant, $V(Ah)^{-1}$ | $Q_r$ | reversible heat energy, $Q_{revesibe}$, W·s |
| $Q_0$ | maximum battery capacity, Ah | $Q_{tab}$ | heat energy of connect tab, W·s |
| $I$ | battery current, A | $R_A$ | anode tab resistance, $\Omega$ |
| $I^*$ | low frequency filtered current, A | $R_C$ | cathode tab resistance, $\Omega$ |
| $A$ | exponential voltage, V | $Q_b$ | current battery capacity, Ah |
| $B$ | exponential zone time constant, $Ah^{-1}$ | $m$ | maximum cycle period at $Q_{ir\_m}$ |
| $T_n$ | nominal ambient temperature, K | $Q_{ir\_m}$ | maximum irreversible energy for 1 cycle |
| $T_i$ | cell internal temperature, K | $Q_{ir\_k}$ | current irreversible energy for 1 cycle |
| $T_a$ | ambient temperature, K | $N_a$ | actual cycle time |
| $\beta$ | Arrhenius rate constant for $R_i$ | $N_p$ | predicted cycle time |

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
