# Peer review of "Novel Practical Life Cycle Prediction Method by Entropy Estimation of Li-Ion Battery"

_electronics, doi:10.3390/electronics10040487_

Round 1

Reviewer 1 Report

Page 5, line 159: the term "species" is unclear in this context.

Page 5, line 164:  "A higher charge and discharge cycle led to faster....discharge rate?" Confusing, simplify what is being stated here?

Page 11, line 293: the process flow diagram is helpful, but an example of the calculations used to predict cycle life (possibly in a supplemental) would be useful for general understanding. 

Page 11, line 294: Please clarify what equations are being used to predict residual lifetime.

Table 3 and Figure 12:  The values in the table do not match the values read off the graphs. Table 3: Predicted is always less than actual.  Graph 12: Predicted  always greater than actual.

Author Response

Thank you for your kind and valuable answer.

I got your opinion, we have added and revised descriptions to reflect your comments requiring revisions.

Page 5, line 159: the term "species" is unclear in this context.

  • -We modified species to samples.

Page 5, line 164: "A higher charge and discharge cycle led to faster....discharge rate?" Confusing, simplify what is being stated here?

  • -I'm sorry. It was explained in a way that I don't understand, too. I explained it a little more understandable on page 5, lines 170-174.

Page 11, line 293: the process flow diagram is helpful, but an example of the calculations used to predict cycle life (possibly in a supplemental) would be useful for general understanding.

  • We attached the calculation formula data in the supplemental.

Page 11, line 294: Please clarify what equations are being used to predict residual lifetime.

  • We clarified by specifying the formula number what equations are being used to predict residual lifetime.

Table 3 and Figure 12: The values in the table do not match the values read off the graphs. Table 3: Predicted is always less than actual. Graph 12: Predicted always greater than actual.

  • Table values and graphs are no different. We apologize for the confusion as we didn't organize the tables in order of their lifetime.
  • Considering the safety factor of the battery system, the minimum Qir_m was chosen as shown in Equation 18 to reduce the error for relationships that have not yet been identified. Our researchers know that irreversible heat energy increases with lifetime, as shown in Figure 6. We are working on this, but research is still insufficient to generalize. I will explain this further in the conclusion.

Once again, thank you for your valuable review.

Reviewer 2 Report

In my opinion, the provided manuscript is on high quality. Only minor changes are required. 

Section 1. Introduction

Line 23: the expression ’best performing’ should be changed to more appropriate.

Line 28:  a reference should be added to the provided information about the batteries that are developed.

Line 48-49 a reference should be added to the information about the batteries capacity.

Section 2

All symbols used in the equations should be explained in the text.

Line 87 and 90: the abbreviation OCV and ESS should be explained.

Section 3

Lines 289-290 ‘’Using this method, four points can be found, and the area is given by the area formula’’ – What does it mean? Please be more precise.

Section 4

Line 356:  The follow sentences should be rewritten.  ‘’It should be so, but I think it will be a way of higher accuracy. We will solve this through further research.’’ The personal statements should be avoided in scientific articles.

Authors Contributions – the contribution of both authors should be stated in this part. Please check the Guidelines for Authors in Electronics journal.

Author Response

Thank you for your kind and valuable answer.

I got your opinion, we have added and revised descriptions to reflect your comments requiring revisions.

Section 1. Introduction

Line 23: the expression ’best performing’ should be changed to more appropriate.

  • We modified ’best performing’ to ‘excellent energy density’.
  • (Could you have any other recommended expressions?)

Line 28: a reference should be added to the provided information about the batteries that are developed.

  • It is organized based on the seminars and technology conference that I attended, and there is no specific reference.

Line 48-49 a reference should be added to the information about the batteries capacity.

  •  Added reference.

Section 2

All symbols used in the equations should be explained in the text.

  •  All symbols used in the equations were explained in Nomenclature in the text.

Line 87 and 90: the abbreviation OCV and ESS should be explained.

  • I explained the abbreviation you pointed out.

Section 3

Lines 289-290 ‘’Using this method, four points can be found, and the area is given by the area formula’’ – What does it mean? Please be more precise.

  • I explained it a little more specifically on lines 293-295.

Section 4

Line 356: The follow sentences should be rewritten. ‘’It should be so, but I think it will be a way of higher accuracy. We will solve this through further research.’’ The personal statements should be avoided in scientific articles.

  • Other reviewers also pointed out, and I have corrected that part in the conclusion.

Authors Contributions – the contribution of both authors should be stated in this part. Please check the Guidelines for Authors in Electronics journal.

  •  It's corrected.

Once again, thank you for your valuable review.

Reviewer 3 Report

In this manuscript, the authors report the estimation of the state of health for lithium-ion batteries which is based on the entropy of the battery. They report the physico-mathematical derivation of the phenomenon and an algorithm to integrate the charge-voltage curves for a battery. Also, they applied the method to a standard battery which verified their assumptions.

The work is interesting and in general, well written.

Detailed comments

  1. Most of the terms in eq 1 to 6 are not explained in the text. Please provide a short explanation for all of them.

  2. What is Qtab in eq 9?

  3. From eq 11 it is clear that the derivation is valid only for constant current. In fact, the current is taken out of the time integral as a constant. The authors should stress and elaborate on this point.

  4. What is the effect of a changing current profile on this methodology?

  5. What is the “DOD”?

  6. It is not clear to me why a simple integration of the charge/discharge curve can be done? The method proposed introduces large errors compared to a simple numerical integration. Also, the proposed method shows the maximal errors when the SOC is pushed toward the limits (0 and 100 %), but these are also some of the conditions which will deteriorate faster the state of health of the battery.

  7. I do not understand the sentence at line 154-155. What is an “A Battery”?

  8. There are also some problems with the line 283 and line 355-357. I do not understand what the authors want to express here.

  9. The number of references to back up and compare the findings and the results is quite limited.

Author Response

Thank you for your kind and valuable answer.

I got your opinion, we have added and revised descriptions to reflect your comments requiring revisions.

Detailed comments

Most of the terms in eq 1 to 6 are not explained in the text. Please provide a short explanation for all of them.

  • All symbols used in the equations were explained in Nomenclature in the text.

What is Qtab in eq 9?

  • Qtab is the heat energy of connecting tab, W∙s in units. Please refer to Nomenclature

From eq 11 it is clear that the derivation is valid only for constant current. In fact, the current is taken out of the time integral as a constant. The authors should stress and elaborate on this point.

  • That's a very accurate and good point out. Actually, since the battery system does not use only constant current, the method of calculating irreversible energy using Equation 11 as in the previous studies includes a time factor, and it was likely to occur many errors.
  • However, we showed that the time component can be substituted with the SOC component, as suggested in Equations 12-15, and we emphasized that the irreversible energy can be effectively calculated through this.
  • Of course, the method of estimating SOC must be accurate. But, as you know, there are many ways to estimate SOC, and it is becoming more accurate, and given the fact that BMS(Battery Management System) to which these techniques are applied are commercialized, I would like to emphasize that our proposed method is very effective in the future.

What is the effect of a changing current profile on this methodology?

  • Regardless of the change in the current profile, the proposed method allows high lifetime prediction. In this paper, we have informed that it affects the lifespan as the irreversible energy used varies depending on the amplitude of the charge/discharge current and the DOD.

What is the “DOD”?

  • DOD is discharge of depth. This was explained on page 2, line 57.

It is not clear to me why a simple integration of the charge/discharge curve can be done? The method proposed introduces large errors compared to a simple numerical integration. Also, the proposed method shows the maximal errors when the SOC is pushed toward the limits (0 and 100 %), but these are also some of the conditions which will deteriorate faster the state of health of the battery.

  •  I will explain in two aspects. First, it is about practicality. The integral method can be very accurate in continuous calculations. However, the real system is a discrete system. For a discrete system to be continuous, the sampling time must be extremely short, which means the main processor must be very performant. Otherwise, discrete integration systems can contain very large errors due to transient responses and cumulative errors. This is the same as the error that occurs when linearly interpolating a nonlinear signal.
  • Also, our proposed method is that the data is not continuously gathered and computed, it is computed only by event occurrence. Like a processor interrupt. Thus, it is possible to eliminate the influence of irregular factors such as transient response and natural discharge. This can simplify the methodology of estimating the lifetime. It's like that when we drive a car, regardless of accelerating, idling, or driving patterns, we can know the remaining mileage by looking at the fuel gauge.
  • Second, it is the actual usage range of the battery. As the reviewer said, near 0% and 100%, it shortens the lifespan and has a factor that damages the material inside the battery, so most battery systems actually have a range of 20 to 80%. Even if you use a little more, it does not exceed 10 to 90%. Considering these, I would like to say that our method is very effective in predicting lifespan.

I do not understand the sentence at line 154-155. What is an “A Battery”?

  •  It means company A. Because it is not possible to mention the name of a specific company.

There are also some problems with the line 283 and line 355-357. I do not understand what the authors want to express here.

  • Other reviewers also pointed out, and I have corrected that part in the conclusion.

The number of references to back up and compare the findings and the results is quite limited.

  • There are a lot of references, but I've written all of the direct references for writing this paper.

Once again, thank you for your valuable review.

Round 2

Reviewer 3 Report

Thank you for the answers